# Green Synthesis of Zinc Oxide Nanoparticles Using Aqueous Extract of *Pavonia zeylanica* to Mediate Photocatalytic Degradation of Methylene Blue: Studies on Reaction Kinetics, Reusability and Mineralization

**DOI:** 10.3390/ijms26104739

**Published:** 2025-05-15

**Authors:** Dhananjay Purushotham, Abhilash Mavinakere Ramesh, Divakara Shetty Thimmappa, Nataraj Kalegowda, Gowtham Hittanahallikoppal Gajendramurthy, Shiva Prasad Kollur, Murali Mahadevamurthy

**Affiliations:** 1Department of Studies in Materials Science, University of Mysore, Manasagangotri, Mysuru 570006, Karnataka, India; dhananjaykp361@gmail.com; 2Department of Studies in Environmental Science, University of Mysore, Manasagangotri, Mysuru 570006, Karnataka, India; abhilash@envsci.uni-mysore.ac.in; 3Department of Botany, Srisaila Jagadguru Vageesha Panditaradhya College, Harihar 577601, Karnataka, India; orchids3804@gmail.com; 4Department of Studies in Botany, University of Mysore, Manasagangotri, Mysuru 570006, Karnataka, India; knataraj922@gmail.com; 5Department of Studies and Research in Food Science and Nutrition, Karnataka State Open University (KSOU), Mysuru 570006, Karnataka, India; gajendramurthygowtham@gmail.com; 6School of Physical Sciences, Amrita Vishwa Vidyapeetham, Mysuru Campus, Mysuru 570026, Karnataka, India; shivachemist@gmail.com

**Keywords:** photocatalysis, methylene blue, *Pavonia zeylanica*, mineralization, kinetics

## Abstract

Nanoparticles (especially zinc and titanium oxide) have been found to be effective in photodegrading pollutants (organic/inorganic) from industrial wastewater. Presently, this study aimed at biosynthesizing zinc oxide nanoparticles (ZnO-NPs) from the leaf extract of *Pavonia zeylanica*, a plant with significant medical value, and evaluating their photocatalytic properties against methylene blue (MB), an azo dye (100 mg L^−1^, pH 7), using solar irradiation, along with the measurement of their reusability and mineralization efficiency. The characterization of the Pz-ZnO-NPs showed an absorbance peak at 313 nm, with a bandgap value of 3.04 eV and a size of 19.58 nm. This study’s results show that the synthesized Pz-ZnO-NPs, upon treatment with MB dye after 2 h of solar irradiation, showed an 89.32% degradation, which was concentration-dependent and followed pseudo-first-order kinetics. The reusability studies indicated that the Pz-ZnO-NPs were able to degrade MB dye after five repeated cycles of its usage. The structural composition of the Pz-ZnO-NPs evaluated by XRD showed that the peak position stayed constant. Nevertheless, the peak intensity dropped, indicating that the ZnO-NPs’ crystal structure was unaffected. Furthermore, advanced oxidation process studies, which included an evaluation of COD and TOC, revealed that both the contents decreased significantly during the photocatalysis process, wherein the electron-rich organic dyes were converted to nontoxic products through mineralization.

## 1. Introduction

Nanotechnology is the process of designing, characterizing, manufacturing, and implementing structures, devices, and systems at the nanometer scale while controlling their shape and size. Nanoparticles are typically 100 nm or smaller and exhibit unique features compared to bigger particles of the same substance [1]. Nanoparticles can be derived in two ways: bottom-up and top-down techniques, and although physical and chemical processes offer high-yield production, they require reductants and stabilizing components that are hazardous for the environment, produce an extensive amount of pollutants, and are expensive and time-consuming [2]. Currently, biological synthesis approaches, often called “green synthesis”, for NPs are gaining popularity due to them using only natural, non-harmful, naturally occurring components at low cost, and they are also environmentally safe [3,4]. This approach attempts to reduce the use of chemicals to develop an environmentally friendly route for manufacturing nanoparticles that possess diverse biological properties with photocatalytic efficiency [5].

Zinc oxide nanoparticles (ZnO-NPs) are garnering significant interest in various scientific fields owing to their unique features (*viz.*, physical, chemical, and biological properties) [6,7]. ZnO-NPs are remarkable semiconductors with a substantial surface area and redox activity and various optical and electrical properties, making them a promising option of significance [8]. These materials’ UV scattering ability makes them valuable for solar cells, UV detectors, gas sensors, and sunscreen formulations [9,10]. Consequently, it is essential to identify plants that can be used for synthesizing ZnO-NPs with the ability to photodegrade dyes as an alternative to available photodegradation agents.

The most significant feature of nanotechnology is the production of metals and their oxide forms of nanoparticles by chemical, physical, or green techniques. The chemical synthesis method uses costly, toxic, and hazardous chemicals, whereas the green technique uses non-harmful, naturally occurring components [5,11]. Green synthesis uses natural, affordable, and environmentally favorable materials, which lowers the production of pollutants and produces high-purity nanoparticles, benefiting both the environment and human health, and it is thereby deemed preferable to chemical synthesis [12,13]. Bioactive phytoconstituents within the plant extracts are thought to have a significant role in lowering, regulating, and stabilizing the utilized metallic ions [14,15]. To date, there are many reports on the utilization of plants as a whole or their parts for synthesizing nanoparticles of different types that can photodegrade many different azo dyes, including methylene blue. Accordingly, ZnO-NPs synthesized using leaves (*Anisochilus carnosus*) [16], stem (*Mussaenda frondosa*) [17], root (*Zingiber officinale*) [18], rhizome (*Rheum turketanicum*) [19], flower (*Panax ginseng*) [20], fruit (*Hippophaer hamnoides*) [21], and fruit pulp (*Aegle marmelos*) [22] have been found to be effective in photodegrading methylene blue dye.

The species *Pavonia* (Malvaceae) grows around the world, including in Africa, Asia, and South America, and it is often used to treat diabetes, gonorrhea, malaria, diarrhea, stomach colic, nausea, rheumatism, and edema in traditional medicine. Importantly, scientific publications have proven that the plant possesses antibacterial, antihyperglycemic, antinociceptive, anti-diarrheal, anti-inflammatory, etc., properties [23,24]. Currently, several bioactive components have been isolated from this medicinal herb, including flavonoids, phenylethyl glycosides, lignans, coumarins, and triglycerides. *Pavonia zeylanica* L. (Malvaceae) is a shrubby herb that is known for its many biological properties [25,26], and it was used during this study for synthesizing ZnO-NPs. While the benefits of green synthesis, which deals with adhering to green chemistry principles and converting waste into beneficial products, are well recognized, there is a scarcity of research that has specifically investigated the potential aqueous solution extracts of *P. zeylanica* leaves for synthesizing ZnO-NPs for safer and more sustainable applications. Furthermore, the plant has not been utilized before to synthesize any metal or metal oxide nanoparticles for photocatalytic applications. Therefore, research was carried out to synthesize ZnO-NPs from *P. zeylanica* and to analyze their photocatalytic ability against the azo dye MB.

## 2. Results and Discussion

### 2.1. Structural and Morphological Characterization of Green-Synthesized Pz-ZnO-NPs

During this study, the utilization of *P. zeylanica* aqueous leaf extract and zinc nitrate hexahydrate through a hydro-thermal method yielded ZnO-NPs (Pz-ZnO-NPs),which were further subjected to structural and morphological characterization. It is well known that biosynthesized ZnO-NPs have a UV–vis absorption spectrum ranging from 280 to 400 nm [27], and a UV–vis spectra peak at 313 nm was noticed for the Pz-ZnO-NPs during this study (Figure 1A). Also, the Pz-ZnO-NPs showed a blue-shifted absorption edge with a band gap value of 3.04 eV (Figure 1B), corresponding to the near-UV region, thereby indicating a significant quantum confinement impact. Likewise, this study’s results align with many other findings, wherein plant-derived ZnO-NPs had an absorption peak between 280 and 400 nm [6,28,29], with a band gap energy ranging from 2.5 to 3.5 eV [30]. However, the UV–vis absorption spectrum (Figure 1A) showed significant tailing into the visible region (400–500 nm). This can be explained by the presence of intrinsic defects such as oxygen vacancies (V_O), zinc interstitials (Zn_i), and surface hydroxyl groups, which created localized energy levels within the bandgap and enabled sub-bandgap visible light absorption. Additionally, phytochemicals from plant extracts may passivate surface states and induce band-tail effects [31]. Therefore, visible light can excite electrons from these defect states to the conduction band, supporting photocatalytic activity under visible light, consistent with other green-synthesized ZnO systems. These defect states allow for the absorption of lower-energy visible photons, facilitating electronic transitions that would otherwise be prevented in a perfect crystal structure. Additionally, surface states resulting from the high surface-to-volume ratio of nanoparticles and possible impurities from the plant-mediated synthesis process further contribute to the visible light absorption. The observed blue-shifted absorption edge, indicative of quantum confinement effects, can also modify the electronic structure, enabling enhanced absorption in the visible region. Thus, visible light can indeed provide additional useful energy to drive photocatalytic reactions by exciting electrons via defect- or surface-mediated transitions, enhancing photocatalytic efficiency under broader light spectra. The results obtained indicate favorable optical features, which may be exploited in several applications, including catalysis, sensing, and energy conversion, as also reported by Raha and Ahmaruzzaman [7].

Figure 1C shows the X-ray diffraction (XRD) patterns of the green-synthesized Pz-ZnO-NPs, wherein the peaks were identified using JCPDS No. 01-079-0206. The Pz-ZnO-NPs diffractograms were consistent with those of the ZnO nanoparticles. The existence of ZnO in the biosynthesized Pz-ZnO-NPs was confirmed by the distinct peaks between 30 and 40 2 theta (2θ) angles *viz.* 31.68, 34.36, and 36.22, which corresponded to the 100, 002, and 101 Bragg reflections of the face-centered cubic structures. The Debye–Scherrer formula, which was applied to determine the size of the Pz-ZnO-NPs, revealed that the particles were ~19.58 nm in size (Appendix A). The sharper XRD peaks might have been caused by the facile migration of Zn+ and O- atoms within the lattice and their eventual occupancy at the vacant sites, allowing for a considerable decrease in the lattice due to annealing [32,33].

The FT-IR spectra of the *P. zeylanica* aqueous extract and the Pz-ZnO-NPs derived from it are shown in Figure 1D. The major bands at 3279.68 cm^−1^ corresponded to the asymmetric wide band of NH2, while the 2348.28 cm^−1^ and 2098.37 cm^−1^ bands corresponded to primary and secondary amide resonance, respectively. Main and secondary amide C=O stretching (CONH_2_) bands appeared at 1609.36 cm^−1^, and an O-H deformation vibration band was discovered at 1398.87 cm^−1^, whereas a C-O wider vibration band was noted at 1018.71 cm^−1^ in the plant extract. This was due to the influence of phytochemicals in the plant acting as a reducing agent and decreasing the content of zinc acetate salt, and there was also a strong band at 547.30 cm^−1^, which indicates ZnO-NP production. The bioactive moieties discovered as surfactants were attached to the surface of the ZnO-NPs and stabilized via electrostatic stabilization, indicating that the aqueous extract of *P. zeylanica* could decrease and stabilize the ZnO-NPs, and these findings support the chemical stability of the Pz-ZnO-NPs. Similarly, FT-IR examination of the ZnO-NPs formulated using plant extracts revealed an absorption peak in the range from 400 to 500 cm^−1^ [34]. During the synthesis of ZnO-NPs using plant extracts, the phytoconstituents/secondary metabolites of the plants actively take part as bioreductants by efficiently chelating and acting as capping agents to aid in the stability and to regulate the crystal development of the synthesized particles [35].

The surface morphology of the PZ-ZnO-NPs was evaluated through scanning electron microscopy (SEM), and the SEM micrograph of the Pz-ZnO-NPs (Figure 2A) demonstrates that the biosynthesized Pz-ZnO-NPs exhibited considerable particle aggregation, resulting in a non-uniform size distribution. The aggregation was mainly attributed to the high surface energy, polarity, and electrostatic interactions characteristic of ZnO nanoparticles, especially those synthesized by green routes, where naturally occurring phytochemicals act as capping and stabilizing agents, which is in agreement with the findings of Pillai et al. [36]. To further establish the elemental properties of the manufactured particles, EDX analysis was performed, wherein peaks for Zn and O elements were observed (Figure 2B). The insert table in Figure 2B shows the weight and atomic percentages of zinc (58.2%) and oxygen (23.2%), indicating that the synthesized photocatalyst mainly consisted of Zn and O, and this proved the presence of zinc in an oxide form. Similarly, Zn and O signals were discovered between 0–2 KeV, and two Zn signals were acquired between 7–10 KeV, which is consistent with findings of Chaudhuri and Malodia [37] and Barzinjy and Azeez [38]. The strong Cl signal observed in the EDS spectrum (Figure 2B) likely originated from residual chloride ions (Cl^−^) present due to the precursors or from the plant extract constituents used during the biosynthesis process. However, this does not necessarily indicate the formation of ZnCl_2_ impurities in the final product, as no additional crystalline phases related to ZnCl_2_ were detected in the XRD analysis (Figure 1C). The absence of characteristic peaks of ZnCl_2_ in the XRD pattern suggests that chloride was either weakly adsorbed on the nanoparticle surface or that it existed in amorphous, non-crystalline forms rather than being incorporated as a separate phase [38]. The presence of Cl^−^ ions on the surface of the ZnO nanoparticles could influence their photocatalytic performance in two ways. Firstly, Cl^−^ ions may slightly passivate surface defects or trap photo-generated charge carriers, thereby potentially affecting the efficiency of photocatalytic reactions. On the other hand, certain studies suggest that surface chloride species can sometimes promote photocatalytic oxidation reactions by facilitating charge separation and suppressing electron–hole recombination, thus enhancing photocatalytic activity under specific conditions [7,14].

The N_2_ adsorption and desorption technique was employed to determine the surface characteristics and porosity of the Pz-ZnO-NPs, and Figure 3 illustrates the N_2_ adsorption/desorption curves. The findings revealed that the Pz-ZnO-NPs exhibited a hysteresis loop of H_3_, suggesting the presence of a mesoporous structure in the manufactured samples with a specific surface area (SBET) of 23.42 m^2^ g^−1^. Likewise, the Pz-ZnO-NPs had a total pore volume of 0.001261 cm^3^ g^−1^ (p/po = 0.906), with an 8.04 nm mean pore diameter. The BET/BJH study shed light on the fundamental features of the materials, demonstrating large surface areas, constant pore volumes, and a wide range of pore diameters, which have tremendous promise for catalytic applications [39,40].

### 2.2. Photocatalytic Degradation (PD) of MB by Pz-ZnO-NPs

The PD of azo dye MB was studied under simulated solar light irradiation, and Figure 4 depicts the dye degradation rates achieved by the photocatalyst used at different concentrations. Throughout the adsorption and desorption process in the dark, dye degradation was not noticed; however, upon exposure to solar irradiation, a significant shift in the effectiveness of the Pz-ZnO-NPs in degrading the MB dye was observed. The overall pattern indicates that the MB dye degradation rates increased with the increase in the photocatalyst loading (Figure 4A). The photodegradation ability of the photocatalyst Pz-ZnO-NPs increased linearly with the exposure duration (120 min) and concentration (2 g L^−1^), with a maximum degradation potential of 89.32% (Figure 4B; Appendix A). The inference is that increasing the loading of the Pz-ZnO-NPs caused more optical absorption in the visible region, which is supported by the UV–vis measurements and estimated absorption wavelengths for different loadings. In addition, this study’s results followed pseudo-first-order photodegradation kinetics, with a best linear fit offering a rate constant of 0.016 min^−1^ (Figure 4C) and an adsorption capacity at equilibrium (qe) of 0.9548 mg g^−1^ (Figure 4D). From recent findings, it may be observed that kinetics following a best linear fit plot can be seen as the rate law governing the photodegradation process [40]. There is substantial support that ZnO-NPs possesses higher reactive groups, which effectively generate hydrogen peroxide and have greater reaction rates that assist in the PD of dyes. This study’s findings corroborate the same, as the photocatalyst Pz-ZnO-NPs offered an increased degradation potential as the catalyst dose increased. Solar irradiation excites electron–hole pairs (e^−^/h^+^), which are formed from ZnO-NPsfrom the valence band to the conduction band, resulting in a photogenerated electron–hole pair (e^−^/h^+^) that subsequently results in the formation of •OH radicals and H_2_O_2_, which contribute to the breakdown of MB dyes through the formation of H_2_O and CO_2_ [41]. The crystallinity, size, and surface area of nanoparticles are deemed to be important factors that affect the photodegradable potential of the synthesized particles, and the green-synthesized Pz-ZnO-NPs had a highly crystalline nature, smaller size, and high surface area, which was validated through XRD, bandgap energy, and BET studies, thereby authenticating the higher photodegradation ability of the synthesized particles. Additionally, these characteristic features of the particles help improve the reactivity and adsorption capacity of dyes by providing more reactive sites for the reaction [42,43]. The valence bands of electron–hole pairs oxidize and interact with H_2_O, producing highly oxidized hydroxyl (•OH^−^) radicals [42]. If the bandgap is wide, the electron–hole pairs recombine back into the VB, and the produced radicals attack the pollutant and degrade the dye, leaving only ecologically benign compounds.

### 2.3. Reliability and Structural Durability of Pz-ZnO-NPs

For investigating the reusability and photostability of the Pz-ZnO-NPs for MB dye photodegradation, cycling tests were carried out utilizing an oven drying technique at 100 °C between reaction cycles [42]. After five successive photodecomposition cycles, the catalytic effectiveness of the Pz-ZnO-NPs remained unchanged. The PD ability for five consecutive experimentations resulted in a somewhat lowered dye degradation from 89.32% to 82.82% (Appendix A), showing that the biosynthesized Pz-ZnO-NPs were more durable and recyclable for MB dye degradation (Figure 5A). The results are in line with those of Khan et al. [44] and SaadAlgarni et al. [45], wherein ZnO-NPs biofabricated using plants were effective even after a recyclability study with up to five cycles for rhodamine B, methylene blue, and congo red degradation, respectively. In addition, the crystalline/structural stability of the Pz-ZnO-NPs after each cycle of photodegradation was investigated by employing XRD analysis (Figure 5B). The study results show that the peak intensity decreased, but the position of the peak stayed constant, entailing that the crystal structure of the ZnO-NPs remained unaffected, which aligns with the results obtained by Singh et al. [46] and Kader et al. [47].

### 2.4. Mineralization Properties of Pz-ZnO-NPs

Chemical oxygen demand (COD) is a better and more well-known measure for determining the intensity of oxidizable pollutants in contaminated water samples [42]. During this study, the COD of the MB dye solution with and without the photocatalyst (Pz-ZnO-NPs) before and after irradiation was calculated, and the COD values were found to be 91.26 mg L^−1^ and 18.92 mg L^−1^, respectively (Figure 5C; Appendix A). The decrease in the COD value of the degraded solution confirms the eradication of the dye molecule in conjunction with the removal of color, indicating the efficacy of the Pz-ZnO-NPs. The findings are consistent with Shinde et al. [48], in which plant-derived ZnO-NPs efficiently degraded the dye solution upon solar irradiation. In addition, the MB dye solution’s TOC was determined after exposure to direct sunshine. The organic carbon contents decreased with the irradiation time in the MB dye solution upon TOC measurement after the addition of the photocatalyst and were found to be 94.12 mg L^−1^ and 25.42 mg L^−1^, respectively (Figure 5D). Likewise, Venkatesan et al. [49,50] noted that bio-fabricated ZnO-NPs from *Solanum trilobatum* and *Citrus limetta* were able to mineralize COD and TOC, respectively, after solar irradiation. The results of this investigation showed that COD and TOC were reduced upon treatment with Pz-ZnO-NPs in the MB dye solution after photodegradation. In addition, ZnO-NPs are known to generate extremely reactive free radicals that break down electron-rich organic dyes into harmless compounds (e.g., CO_2_ and H_2_O) through mineralization [42]. The PD study’s results followed an advanced oxidation process (AOP), wherein the biodegradability process resulted in the production of hydroxyl radicals, which, in the presence of catalytic molecules (here, Pz-ZnO-NPs) were broken down without producing any secondary pollutants like hydrogen peroxide and leaving only ecologically benign compounds such as H_2_O, CO_2_, and less toxic minerals [51].

### 2.5. Photocatalytic Mechanism

Upon exposure to visible light, Pz-ZnO-NPs absorb photons that possess energy equivalent to or surpassing their inherent bandgap energy, leading to the excitation of electrons from the valence band to the conduction band, thereby engendering the formation of positively charged holes within the valence band [52]. These photogenerated charge carriers, specifically electrons and holes, assume a pivotal role in the generation of reactive oxygen species, which are instrumental in the degradation of organic pollutants. The holes (h^+^_vb) generated within the valence band instigate the oxidation of water molecules (H_2_O) or hydroxide ions (OH^−^) adsorbed onto the catalyst surface, culminating in the production of highly reactive hydroxyl radicals (•OH), renowned for their potent oxidizing capabilities [53]. Concurrently, the excited electrons residing in the conduction band facilitate the reduction of dissolved molecular oxygen, culminating in the formation of superoxide radicals, which subsequently transform into other reactive species, encompassing hydroperoxyl radicals and additional hydroxyl radicals [54]. These ROS attack the dye molecules, breaking them into harmless end products like carbon dioxide (CO_2_), water (H_2_O), and other mineral acids. The use of Pz-ZnO-NPs enhances photocatalytic performance by improving light absorption, stabilizing particle dispersion, and facilitating charge carrier separation due to the presence of bioactive phytochemicals. The dye degradation mechanism is described in Figure 6. A schematic illustration of the PD by the biosynthesized catalyst Pz-ZnO-NPs is given in Figure 7. Furthermore, developing and using efficient solar-activated materials is essential when aiming to not leave any contaminants due to environmental contamination.

## 3. Materials and Methods

### 3.1. Green Synthesis of ZnO-NPs from Pavonia zeylanica

*Pavonia zeylanica* plant leaves were obtained from Mysuru City (Karnataka State, India), which were washed first and then blot-dried before being pulverized using double-distilled water (1:10 *w*/*v*). The resultant mixture was shaken in a rotating shaker (2 h at 100 rpm) and filtered using Whatman No. 1 filter paper [27]. Zinc nitrate hexahydrate was incorporated into the plant extract at 60–80 °C (10:1 *v*/*w*) on a heated magnetic stirrer until the mixture was reduced to 1/10th of the volume. The obtained product was then moved into a crucible and calcined at 300 °C for 3 h, yielding green-synthesized ZnO-NPs that were employed for physicochemical characterization, followed by assessing their photodegradation ability.

### 3.2. Structural and Morphological Characterization of Green-Synthesized Pz-ZnO-NPs

This study analyzed the morphology, size, and chemical composition of the synthesized nanoparticles, and these were evaluated by UV–vis spectroscopy (Labtech, Tamil Nadu, India), X-ray diffraction (XRD; Rigaku Smart Lab, Tokyo, Japan using CuKa radiation), Scanning electron microscopy (SEM; HITACHI S-4800, Tokyo, Japan), Energy-dispersive X-ray spectroscopy (EDS; Zeiss Supra 55VP, Baden-Württemberg, Germany), and Fourier-transform infrared spectroscopy (FT-IR; PerkinElmer Spectrum 1000, Waltham, MA, USA). The Braunauer–Emmett–Teller (BET) study of the Pz-ZnO-NPs was carried out using BEL BELSORP-Mini II (Osaka, Japan)The nanoparticles were analyzed both on their surfaces and in-depth to understand their shape, size, and chemical makeup thoroughly.

### 3.3. Photocatalytic Degradation by Pz-ZnO-NPs

The photocatalytic chemical breakdown of MB dye under an illumination source with varying concentrations of Pz-ZnO-NPs (0–2 g L^−1^) was examined according to the method of Dhananjay et al. [28]. In brief, an MB dye solution (100 mg L^−1^) was prepared using distilled water (pH of 7), and known quantities of Pz-ZnO-NPs were introduced and sonicated for 5 min before incubation on a stirrer for 60 min (dark conditions) to achieve adsorption and desorption equilibrium. The incubated flasks holding the reaction mixture were subjected to direct natural sunlight irradiation (~430–790 THz) under continual agitation to ensure uniform mixing. While the intensity of natural sunlight varies, the trials were conducted under similar daytime conditions to ensure reproducibility. Each reaction mixture was collected every 30 min until 2 h of exposure. The obtained mixture from the reaction was instantly centrifuged for 5 min at a speed of 3000 rpm, and the collected supernatant absorbance was noted at 662 nm to quantify the residual quantity of MB dye.$$Degradation\;(\%)=\frac{C_{0}-C_{t}}{C_{0}}\times 100$$
where C_0_ represents the dye solution’s initial concentration, and C_t_ represents the dye solution’s concentration after the reaction time.

Furthermore, the quantitative kinetics of the increase in photocatalytic activity and the degradation of the MB dye over time were investigated using pseudo-first-order kinetics.−ln (C_t_/C_0_) = k(p) × t
where C_t_ stands for the absorbance of MB at time t, Co stands for the absorbance of MB after dark adsorption, and t stands for the irradiation time. The pragmatic kinetic rate constant, k(p), was determined from the slope of ln(C_t_/C_0_) vs. the irradiation time(t).

### 3.4. Reliability and Structural Durability of Pz-ZnO Nanoparticles

To assess the reusability of the Pz-ZnO-NPs as photocatalysts, photodegradation studies were performed five times by recovering the catalyst each time and applying the same catalyst against a newly prepared MB dye solution (100 mg L^−1^, pH 7). The catalysts were recovered by centrifugation (10,000 rpm for 10 min) and were washed off using DI water to remove any organic impurities that could have caused cross-contamination. The PD performance was determined by assessing the percentage of deterioration after each cycle. XRD spectroscopy was used to assess the photocatalysts’ photochemical stability after each PD cycle.

### 3.5. Mineralization Properties of Pz-ZnO-NPs

The mineralization efficacy of the Pz-ZnO-NPs (photocatalyst) upon treatment with MB dye solution was estimated through evaluation of the chemical oxygen demand (COD) and total organic carbon (TOC) of the MB dye solution before and after photocatalysis treatment, according to the method of Dhananjay et al. [28]. To calculate the TOC for every sample, a9210p On-line TOC Analyzer was utilized.$$\mathrm{COD}\;\mathrm{Removal}\;(\%)=\frac{{COD}_{0}-{COD}_{t}}{{COD}_{0}}\times 100$$$$\mathrm{TOC}\;\mathrm{Removal}\;(\%)=\frac{{TOC}_{0}-{TOC}_{t}}{{TOC}_{0}}\times 100$$
where COD_0_ and TOC_0_ represent the initial MB concentration (mg L^−1^), and C_t_ represents the MB concentration (mg L^−1^) before and after the photocatalysis at different time intervals.

### 3.6. Analytical Statistics

This study involved three replicates of the experiments, and to identify statistically significant results, the mean values were submitted to Tukey’s honest significant difference (HSD) test using SPSS-Inc. 16.0, with an F-value of *p* < 0.05.

## 4. Conclusions

This study successfully synthesized ZnO-NPs using a leaf extract of *P. zeylanica*, and these NPs were used as a photocatalytic material against MB dye at neutral pH and room temperature under solar irradiation. The physico-chemical characterization of the produced photocatalyst Pz-ZnO-NPs was investigated using UV–vis spectroscopy, XRD, SEM, EDS, FT-IR, and BET analyses. The XRD results showed the existence of wurzite shapes with sharp peaks in the Pz-ZnO-NPs, while the SEM examination confirmed the agglomeration of the particles, which assisted in the improved PD of the produced catalyst. In addition, the existence of zinc and oxygen elements was confirmed through EDS analysis, and FT-IR revealed the contribution of phytoconstituents during the synthesis/stabilization process of the Pz-ZnO-NPs. Furthermore, UV–visible spectroscopy revealed that the Pz-ZnO-NPs possessed significant PD ability, where after the breakdown of MB dye after 120 min under solar irradiation, a maximum photodegradation of 89.32% was noted at a concentration of 2 mg L^−1^, which progressed with pseudo-first-order kinetics. The studies on the recyclability and structural stability of the Pz-ZnO-NPs disclosed that the synthesized particles were efficient in degrading 82.82% of the MB dye after five repeated cycles of usage, and the structural composition of the Pz-ZnO-NPs evaluated by XRD entailed that the peak position stayed constant but the intensity decreased, thereby affirming that the crystal structure was not affected. In addition, the COD and TOC tests indicated that the electron-rich organic pigments were transformed into harmless compounds by mineralization.

## Figures and Tables

**Figure 1 ijms-26-04739-f001:**
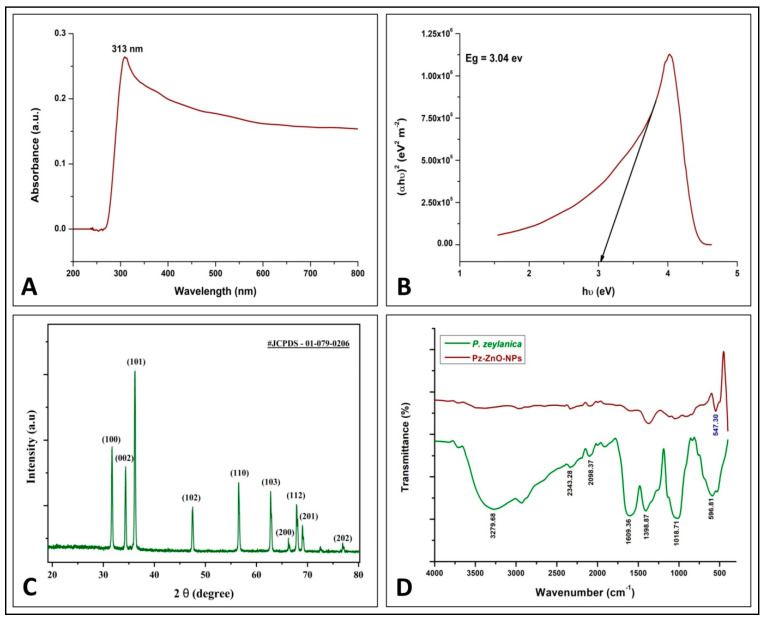
Physico-chemical characterization of Pz-ZnO-NPs. (**A**) UV absorption spectra; (**B**) band gap study; (**C**) powder X-ray diffraction pattern; (**D**) FT-IR spectra ofPz-ZnO-NPs.

**Figure 2 ijms-26-04739-f002:**
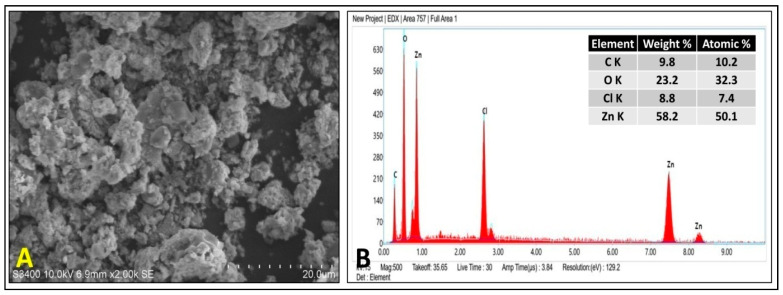
SEM micrograph (**A**) and EDAX spectra (**B**) of Pz-ZnO-NPs.

**Figure 3 ijms-26-04739-f003:**
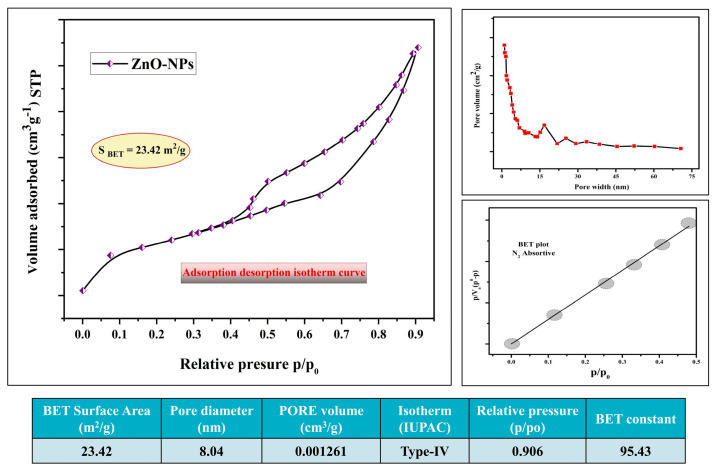
BET surface area studies of Pz-ZnO-NPs.

**Figure 4 ijms-26-04739-f004:**
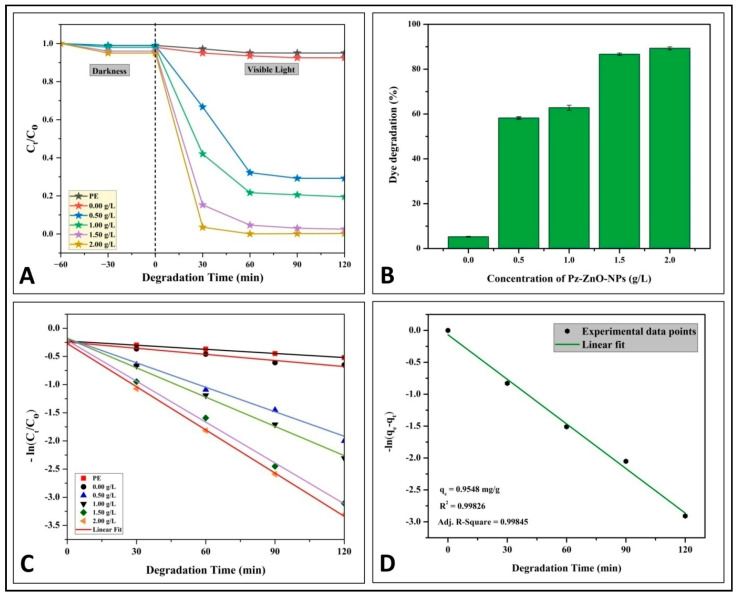
(**A**) Study on photodegradation in the dark and under solar irradiation; (**B**) MB dye photodegradation percentage; (**C**,**D**) investigations of pseudo-first-order kinetics.

**Figure 5 ijms-26-04739-f005:**
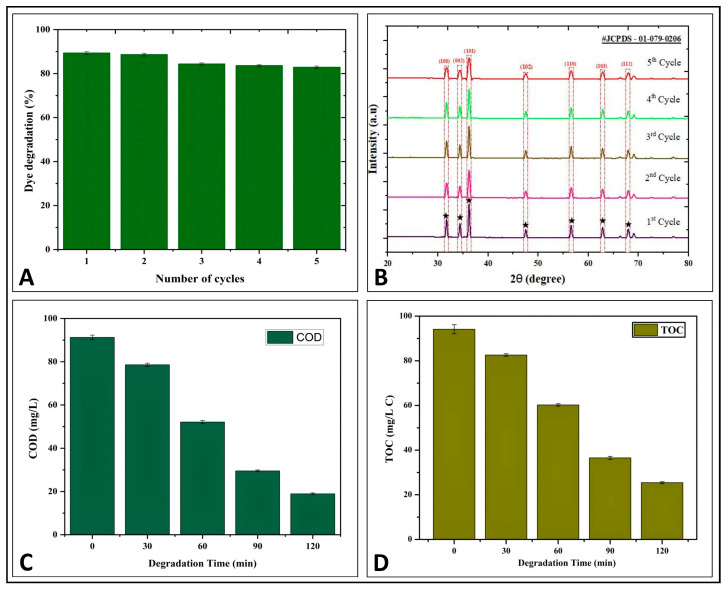
(**A**) Reusability study up to 5th cycle; (**B**) structural stability through XRD study; (**C**) COD study of Pz-ZnO-NPs; (**D**) TOC study of Pz-ZnO-NPs.

**Figure 6 ijms-26-04739-f006:**
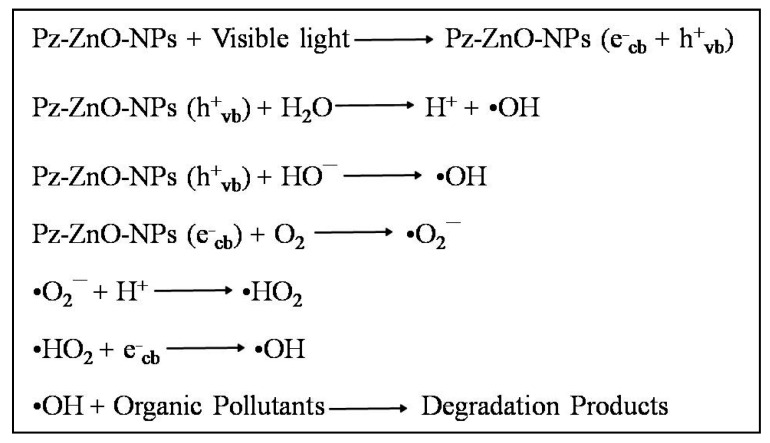
Step-by-step photodegradation mechanism of MB dye by Pz-ZnO-NPs.

**Figure 7 ijms-26-04739-f007:**
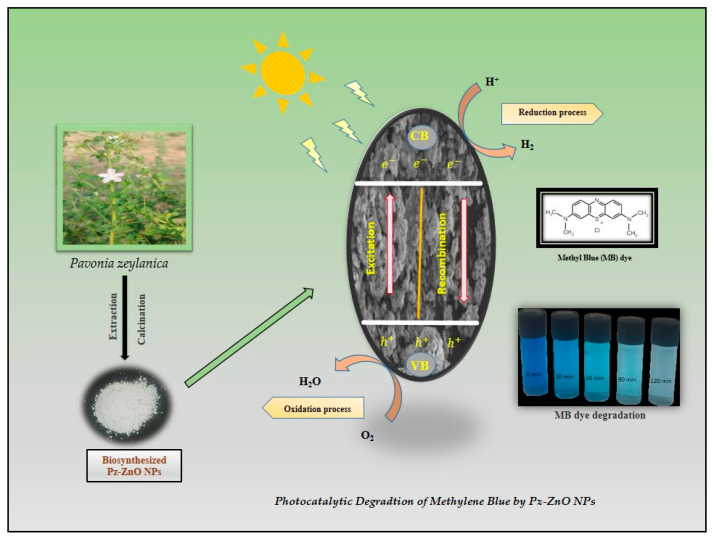
Schematic synthesis and mechanism of photodegradation by Pz-ZnO-NPs.

## Data Availability

All data produced during this work are included in this published article.

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
