# Peer review of "Green Synthesis of Zinc Oxide Nanoparticles Using Aqueous Extract of Pavonia zeylanica to Mediate Photocatalytic Degradation of Methylene Blue: Studies on Reaction Kinetics, Reusability and Mineralization"

_ijms, 2025, doi:10.3390/ijms26104739_

Round 1
Reviewer 1 Report
Comments and Suggestions for Authors
This study presents an eco-friendly approach for synthesizing ZnO-NP photocatalysts toward industrial wastewater treatment, with systematic characterization elucidating their structure-property relationships. Overall, it is interesting and well-organized, but there are some problems that must be dealt with:
1. The reported bandgap of ~3.04 eV (Fig. 1B) appears inconsistent with the notable visible light absorption shown in Fig. 1A. Please clarify what mechanisms enable significant absorption of lower-energy visible light. Can visible light provide additional useful energy to drive photocatalytic reactions?
2. Please discuss the potential impact of severe particle aggregation (Fig. 2A) on the photocatalytic efficiency, and propose strategies (e.g., surfactant modification or ultrasonication) to mitigate aggregation in future studies.
3. A strong Cl signal is observed in the EDS spectrum (Fig. 2B). Does this indicate ZnCl₂ impurities? How might the Cl⁻ ions affect the photocatalytic process? Whether post-treatment like water washing can remove such impurities?
4. Photocatalytic efficiency was tested under natural sunlight, where irradiation intensity varies temporally. The reproducibility of the experiment is questionable. Quantification with radiometers or controlled solar simulators may relieve such concern.
5. To strengthen practical relevance, it is recommended to compare the performance of the new material prepared in this study with typical photocatalytic materials such as commercial ZnO and TiO2. Moreover, in future research, testing in actual wastewater systems containing multiple pollutants or salts is suggested.
6. The quality of figures need to be improved. The font size is too small and the font is inconsistent in some figures. Besides, please check and provide complete captions.
Reviewer 2 Report
Comments and Suggestions for Authors
The manuscript titled "Green Synthesis of Zinc Oxide Nanoparticles (ZnO-NPs) Using Aqueous Extract of Pavonia zeylanica to Mediate Photocatalytic Degradation of Methylene Blue: Studies on Reaction Kinetics, Reusability and Mineralization" presents an interesting approach to the green synthesis of ZnO nanoparticles and their application in photocatalytic degradation. While the topic is relevant and aligns with current research interests, the manuscript requires minor revisions to improve clarity, consistency, and scientific rigor. Specific comments are provided below:
- Please carefully check the manuscript for grammatical errors and spelling mistakes. For example, on line 27, “paek” appears to be a typo, and on line 32, “bye” seems to be incorrect. A thorough proofreading is recommended to ensure clarity and accuracy throughout the text.
- Please provide a supporting reference for the statement: “Furthermore, the plant has not before been utilized to manufacture any form of nanoparticle except for silver nanoparticles with a photocatalytic application.” This claim requires citation to ensure its validity and to acknowledge prior work, if any.
- In Figure 5, the peaks in the XRD pattern should be properly indexed to identify the corresponding crystal planes and phases. This will enhance the clarity and scientific value of the figure.
- Figure 6 appears to resemble a Table of Contents (TOC) graphic. It is unnecessary to include repeated results such as photodegradation, recyclability, and XRD studies in this figure. Please consider streamlining the content to avoid redundancy and improve clarity.
- The mechanism illustrated in Figure 6 is not explained in sufficient detail within the manuscript. Please provide a comprehensive explanation of the proposed mechanism in the main text to enhance the reader’s understanding and support the visual representation.
Round 2
Reviewer 1 Report
Comments and Suggestions for Authors
Sufficient revisions have been made in response to my comments, so I recommend publishing in the current version. Please double check the format of the newly added references to ensure they meet the journal's guidelines
Author Response
Comment 1: Sufficient revisions have been made in response to my comments, so I recommend publishing in the current version. Please double check the format of the newly added references to ensure they meet the journal's guidelines.
Response: We thank the reviewer for their kind acceptance of the submitted MS and also would like to state that we have double checked all the references and ensured that they meet the journal's guidelines.